# Nanocrystal Encapsulation, Release and Application Based on pH-Sensitive Covalent Dynamic Hyperbranched Polymers

**DOI:** 10.3390/polym11121926

**Published:** 2019-11-22

**Authors:** Yunfeng Shi, Gaiying Lei, Linzhu Zhou, Yueyang Li, Xiaoming Zhang, Yujiao Yang, Han Peng, Rui Peng, Huichun Wang, Xiufen Cai, Xinglong Chen, Mengyue Wang, Gang Wang

**Affiliations:** 1School of Chemistry and Chemical Engineering, Anyang Normal University, Anyang 455000, China; 18637818665@163.com (Y.L.); 18337278676@163.com (X.Z.); y15042611565@163.com (Y.Y.); ph18337278759@163.com (H.P.); pr15896849973@163.com (R.P.); Wanghuichun@163.com (H.W.); 18337278435@163.com (X.C.); cxl17152@163.com (X.C.); myw0618@163.com (M.W.); 2Henan Province Key Laboratory of New Optoelectronic Functional Materials, Anyang Normal University, Anyang 455000, China; 3School of Chemistry and Chemical Engineering, Henan Normal University, Xinxiang 453007, China; lgying1006@163.com; 4School of Chemistry and Chemical Engineering, Shanghai Jiao Tong University, 800 Dongchuan Road, Shanghai 200240, China; linzhu.zhou@hotmail.com; 5School of Chemistry and Chemical Engineering, Henan University of Technology, Zhengzhou 450001, China

**Keywords:** nanocrystal encapsulation, release, pH-sensitive covalent dynamic hyperbranched polymer, cell imaging, hydrogenation catalysis

## Abstract

A new strategy for nanocrystal encapsulation, release and application based on pH-sensitive covalent dynamic hyperbranched polymers is described. The covalent dynamic hyperbranched polymers, with multi-arm hydrophobic chains and a hydrophilic hyperbranched poly(amidoamine) (HPAMAM) core connected with pH-sensitive imine bonds (HPAMAM–DA), could encapsulate CdTe quantum dots (QDs) and Au nanoparticles (NPs). Benefiting from its pH response property, CdTe QDs and Au NPs encapsulated by HPAMAM–DA could be released to aqueous phase after imine hydrolysis. The released CdTe/HPAMAM and Au/HPAMAM nanocomposites exhibited excellent biological imaging behavior and high catalytic activities on *p*-nitrophenol hydrogenation, respectively.

## 1. Introduction

Dendritic polymers have three-dimensional structures and many functional groups at terminal. They are of great interest in various applications such as self-assembly [1,2,3,4,5,6,7], control-release of guest molecules [8,9,10,11,12,13], drug and protein delivery [14,15,16,17], gene transfection [18,19,20,21,22], bio-imaging [23], and so on.

Nanocrystal (NC) synthesis and NC encapsulation have been important applications of dendritic polymers over the years [24,25,26]. Until now, dendritic poly(amidoamine)s (PAMAM) [27,28,29,30,31,32], poly(ethylenimine)s (PEI) [20,33,34,35,36,37,38], poly(propylenimine)s (PPI) [39], polyglycerols (PG) [40], poly(amine ester)s (PAE) [41] and their derivatives have often been used as stabilizers for NC synthesis. By preparing or encapsulating nanocrystals (NCs) within dendritic polymers, the properties of NCs (such as optical, electrical and magnetic properties) and the characteristics of dendritic polymers (such as biocompatibility, gene transfection, guest delivery and mechanical property) can be effectively integrated together. The biocompatibility, and optical and magnetic characteristics of NCs could also be tuned using this strategy. In our previous report [42], pH-responsive and double-hydrophilic hyperbranched polymers with acylhydrazone bonds were used to prepare cadmium sulfide (CdS)QDs. In addition, the pH-responsive CdS QDs were applied as a fluorescent probe in acidic lysosomes. However, the low quantum yield and broad fluorescence spectrum of QDs synthesized by these polymers still need to be improved. If QDs with high quantum yield and narrow fluorescence spectrum could be encapsulated by these polymers, these problems could be solved.

For NC synthesis, NCs are synthesized in situ using various stabilizers. Meanwhile, for NC encapsulation, prefabricated NCs are often encapsulated by unimolecular micelles. The NCs are encapsulated in the interior of unimolecular micelles. For NC encapsulation, amphiphilic PAMAM [43] and PEGylated PAMAM [44] have been used for NC encapsulation. However, these dendritic polymers do not have an environmental response, and thus the NCs encapsulated by these polymers cannot be released to another phase, greatly limiting their application. If dendritic polymers with environmental response such as pH, temperature, redox and photo response were used for NC encapsulation, NCs would be endowed with environmental response characteristic. By this strategy, the phase transfer of NCs between organic phase and aqueous phase can be realized, and the optical (such as CdTe quantum dots (QDs)) and magnetic (such as Fe_3_O_4_ NCs) properties of NCs can be tuned during phase transfer. Furthermore, the application of the resulting NCs (such as CdTe QDs) could be extended to not only optoelectronics but also biomedical field (such as bio-imaging, gene transfection).

In this paper, a covalent dynamic hyperbranched polymer with a hyperbranched poly(amidoamine)s (HPAMAM) core and several hydrophobic dodecyl aldehyde (DA) chains connecting with imine bonds (HPAMAM–DA) was constructed. The amphiphilic HPAMAM–DA could encapsulate NCs such as CdTe QDs and Au nanoparticles (NPs), endowing CdTe QDs and Au NPs with pH response property and biocompatibility. Benefiting from the pH response behavior of HPAMAM–DA, CdTe QDs and Au NPs could be released to acidic aqueous phase (pH 5.4) by breaking imine linkage in the HPAMAM–DA. The released CdTe/HPAMAM and Au/HPAMAM nanocomposites were applied in cell imaging and hydrogenation catalysis, respectively.

## 2. Experimental

### 2.1. Materials

Water-soluble CdTe QDs with mercaptopropionic acid as stabilizers (absorption peak 521 nm) were prepared according to reference 43. Linear polyacrylic acid (PAA) oligomer (from Appendix A, *M*w = 1700, *PDI* = 1.40) was synthesized according to reference 45 [45]. Methyl acrylate (99%), ethylenediamine (99%), dodecyl aldehyde (95%), and HAuCl_4_·3H_2_O (99.99%) was bought from Alfa Aesar (Lancashire, UK). Ultrapure water with 18.2 MΩ·cm was used in the whole experiments.

### 2.2. Synthesis

#### 2.2.1. Synthesis of Hyperbranched Poly(amidoamine)s (HPAMAM)

Methyl acrylate (MA, AB type monomer) and ethylenediamine (EDA, Cn type monomer) monomers were applied to synthesize HPAMAM by stepwise polymerization. 0.3 mol EDA and 28 mL methanol were added into a flask. 0.3 mol MA was then added drop by drop while stirring. The reaction was continued for at least 2 days to make sure that all the carbon-carbon double bonds of MA had reacted with amines of EDA by Michael addition. The flask was then fixed onto rotary evaporator to remove all the methanol solvent under vacuum. The resulting intermediate was then heated for 1 h at 60 °C, 2 h at 100 °C, 2 h at 120 °C and 2 h at 140 °C on the rotary evaporator in vacuum, and HPAMAM (from Appendix A, *M*_w_ = 3.8 × 10^3^, *PDI* = 1.27) was gained [33]. Worthy of mention was that an oil pump was used for the reaction at 120 and 140 °C to get a high vacuum degree and high molecular weight of HPAMAM.

^1^H NMR (400 MHz, D_2_O, 298 K) δ: 1.79–2.39 (NH_2_, NH), 2.4–2.52(COCH_2_), 2.53–2.87 (NHCH_2_, NH_2_CH_2_), 2.88–3.39 (NCH_2_).

IR (cm^−1^), 3286 (ν_s NH2_), 3074 (ν_NH_), 2935 (ν_as CH2_), 2835 (ν_s CH2_), 1647 (δ_NH2_), 1547 (δ_NH_, ν_C–N_).

#### 2.2.2. Synthesis of Covalent Dynamic Hyperbranched Polymer with Imine Linkage (HPAMAM–DA)

0.1501 g HPAMAM was dissolved with 50 mL anhydrous ethanol in a 150 mL flask. 0.1191 g dodecyl aldehyde (DA) dissolved in 35 mL anhydrous ethanol was then added into the flask. The system was vigorously stirred and refluxed for two days under argon atmosphere. After cooling to room temperature, vacuum rotatory evaporation and vacuum drying was done to remove all the solvent, thus HPAMAM–DA (from Appendix A, *M*_w_ = 6.0 × 10^3^, *PDI* = 1.05) was obtained. It can be estimated that about 13 DA arms were grafted onto each HPAMAM core.

^1^H NMR (400 MHz, CDCl_3_, 298 K) δ: 0.61–1.04 (CH_3_), 1.05–1.40 ((CH_2_)_10_), 1.98–2.20 (NH_2_, NH), 2.29–2.42 (COCH_2_), 2.43–2.94 (NHCH_2_, NH_2_CH_2_), 2.96–3.65 (NCH_2_), 7.47–7.59 (CONH), 7.89–8.26 (N=CH).

IR (cm^−1^), 3281 (ν_s NH2_), 3072 (ν_NH_), 2960(ν _CH3_), 2921(ν_as CH2_), 2851 (ν_s_
_CH2_), 1653 (ν_C=N_, δ_NH2_), 1563 (δ_NH_, ν_C–N_).

#### 2.2.3. Encapsulation and Release of CdTe QDs Based on HPAMAM–DA Covalent Dynamic Hyperbranched Polymer

300 µL aqueous solutions of CdTe QDs were added to 10 mL chloroform solutions of HPAMAM–DA (0.4 mg/mL). These were gently stirred for one day at 5 °C. The color of aqueous phase faded to colorless under daylight, while greenyellow fluorescence could be seen in the chloroform solution under the ultraviolet lamp, indicating that CdTe QDs had been transferred into chloroform phase by HPAMAM–DA encapsulation. After removing the upper aqueous phase, the CdTe/HPAMAM–DA chloroform phase was diluted to 0.2 mg/mL (according to the concentration of HPAMAM–DA). 10 mL diluted CdTe/HPAMAM–DA chloroform solution was then mixed with 10 mL PAA/HCl (pH 5.4) aqueous solution. After stirring vigorously at 5 °C for 24 h, CdTe QDs were released from chloroform phase to aqueous solution in the form of CdTe/HPAMAM nanocomposites.

#### 2.2.4. Encapsulation and Release of Au NPs Based on HPAMAM–DA Covalent Dynamic Hyperbranched Polymer

10 mg HPAMAM–DA was dissolved in 10 mL chloroform in a 25 mL sample vial. 2 mL HAuCl_4_·3H_2_O aqueous solution (0.75 mM) was adjusted to pH 7 by 1 M NaOH and then added to the sample vial under stirring. Au NPs encapsulated by HPAMAM–DA could be seen in the chloroform phase 24 h later, as evidenced by the claret-red color. After removing the aqueous phase, the concentration of chloroform phase was diluted from 1 mg/mL to 0.4 mg/mL (according to the concentration of HPAMAM–DA). Then 10 mL diluted Au/HPAMAM–DA chloroform solution and 10 mL PAA/HCl aqueous solution (pH 5.4) were added to a sample bottle and vigorously stirred at 5 °C. After 24 h, it could be clearly observed that the color of upper aqueous phase had changed to claret red, indicating that Au NPs had been transferred into aqueous phase.

### 2.3. Cell Viability

Hela cells were seeded into 96-well plates (8000 cells/well) in 200 µL medium and incubated for 24 h. The culture medium was then replaced by 200 µL medium containing series dilutions of HPAMAM or CdTe/HPAMAM at a concentration of 1, 5, 10, 15, 20, 50 and 100 µg/mL, respectively. The Hela cells were cultured for another 24 h. After that, 20 µL 3-(4,5-methylthiazol-2-yl)-2,5-diphenyltetrazolium bromide (MTT) assay stock solution in phosphate buffer solution (PBS) (5 mg/mL) was added to each well and the cells were incubated for 4 h subsequently. After removing the final medium carefully, the remained blue formazan crystals were dissolved in 200 µL dimethyl sulfoxide (DMSO) per well and their absorbance at 490 nm was measured by Elx800 Multilabel counter (PerkinElmer, Waltham, MA, USA).

### 2.4. Application of CdTe/HPAMAM and Au/HPAMAM Nanocomposites

#### 2.4.1. Cell Imaging of CdTe/HPAMAM Nanocomposites

Cell imaging characterization was performed on a confocal laser scanning microscopy. HeLa cells with at a seeding density of 100,000 cells/well were seeded on coverslips in 12-well tissue culture plates and incubated for 24 h. The CdTe/HPAMAM nanocomposites in 200 μL Dulbecco’s modified Eagle’s medium (DMEM) were then introduced and the Hela cells were maintained at 37 °C for 6 h. The cells were washed with PBS and fixed for 30 min with 4% formaldehyde at room temperature. After that, the slides were dipped twice in PBS. The final slides were mounted and measured by a LSM 510META (Carl Zeiss, Jena, Germany).

#### 2.4.2. Hydrogenation Catalysis of Au/HPAMAM Nanocomposites

The hydrogenation catalysis of Au/HPAMAM nanocomposites on *p*-nitrophenol (PNP) was monitored by UV–Vis. 2 mL ultrapure water, 0.75 mL of 0.2 M NaBH_4_ (pH 11) and 0.25 mL of 600 µM PNP aqueous solution (pH 12) were added to a quartz cuvette in sequence. 0.15 mL concentrated Au/HPAMAM solution (pH 11.0) was speedily added into the quartz cuvette. At the same time, UV–Vis data were collected one time per second. The corrected absorbance (after subtracting the background absorbance at 600 nm) at 400 nm (the absorption peak of PNP) was applied to determine the rate constant *k*_app_ by fitting to first-order integrated rate equations.

### 2.5. Measurements

^1^H-NMR spectra were acquired by using a varian Mercuryplus 400 NMR spectrometer (Agilent, Santa Clara, CA, USA). Photographs were done on a SAMSUNG WB150F digital camera (Samsung, Seoul, South Korea). UV–Vis spectra were measured on a Cary 60 spectrophotometer (Agilent, Santa Clara, CA, USA). Photoluminescence spectra were measured on a Cary Eclipse spectrometer (Agilent, Santa Clara, CA, USA) with excitation wavelength at 370 nm. Transmission electron microscopy (TEM) and elemental characterization were conducted on a FEI Tecnai F20 microscope (Thermo Fisher Scientific, Waltham, MA, USA) with an energy-dispersive x-ray spectrometer (EDS, Thermo Fisher Scientific, Waltham, MA, USA) at an accelerating voltage of 200 kV. Dynamic light scattering (DLS) measurements were performed on a Zetasizer Nano-ZS (Malvern Instruments, Malvern, United Kingdom).

## 3. Results and Discussion

HPAMAM were synthesized by Michael-addition and polycondensation using methyl acrylate and ethylenediamine monomers, as shown in Scheme 1a,b. HPAMAM, with many terminal amine groups, could react with dodecyl aldehyde to obtain HPAMAM–DA with imine linkage, as shown in Scheme 1b,c. The chemical structure of HPAMAM–DA was charaterized by NMR. From Figure 1, the chemical shift at 9.5 ppm corresponding to –CHO cannot be seen, implying that –CHO groups from dodecyl aldehyde had been reacted completly with HPAMAM. A new proton signal at 7.97 ppm appears, indicating that imine bonds were formed. Figure 2 shows a comparison of FTIR spectra of HPAMAM and HPAMAM–DA. The N–H stretching vibration of primary and secondary amine groups in HPAMAM locate at 3286 cm^−1^ in Figure 2a, and it shift to 3281 cm^−1^ in Figure 2b for HPAMAM–DA. The bands at 2935 and 2835 cm^−1^ correspond to asymmetric –CH_2_– stretching vibration and symmetric –CH_2_– stretching vibration, respectively. The band at 2960 cm^−1^ in Figure 2b belongs to –CH_3_ stretching vibration, which indicates that DA molecules have been grafted on the HPAMAM. The band at 1647 cm^−1^ in Figure 2a is assigned the bending vibration of primary amines and secondary amines, while it shifted to 1653 cm^-1^ after DA were grafted onto HPAMAM, as shown in Figure 2b.

The amphiphilic and pH responsive HPAMAM–DA, with many hydrophobic chains and a hydrophilic core with many amine groups, could encapsulate water-soluble CdTe QDs or HAuCl_4_ with stirring, as shown in Scheme 1c,d and Figure 3a–d. The amine groups of HPAMAM–DA could chelate with CdTe QDs or HAuCl_4_ molecules and then they capture the CdTe QDs or HAuCl_4_ into the hydrophilic part of HPAMAM–DA. Aqueous HAuCl_4_ encapsulated by HPAMAM–DA were further reduced into Au NPs by the amine groups of HPAMAM–DA, resulting into Au/HPAMAM–DA nanocomposites. The nitrogen atoms in the amino groups of HPAMAM–DA lost electrons to form the oxidized HPAMAM–DA, and Au^3+^ acquired the electrons to be reduced into Au0. After slow aggregation of isolated Au atoms, Au NPs capped by HPAMAM–DA were formed. Benefiting from the pH responsive property of HPAMAM–DA, its imine linkage was broken by PAA/HCl aqueous solution, thus CdTe QDs (or Au NPs) were released in the form of CdTe/HPAMAM (or Au/HPAMAM), as illustrated in Scheme 1d,e and Figure 3e,f. Profiting from the ideal HPAMAM vector, the released CdTe/HPAMAM and Au/HPAMAM should have excellent performance in bio-imaging and hydrogenation catalysis (shown in Scheme 1f,g), respectively.

With an amphiphilic core–shell topological structure, the maximum encapsulation of CdTe QDs by HPAMAM–DA was investigated [43]. Various volumes of aqueous CdTe QDs with same concentration were added to the HPAMAM–DA chloroform solution and gently stirred for 24 h. When the oil-water phase was completely separated, the chloroform solution was filtered through a 220 nm nylon filter and then characterized by UV–Vis spectra which are shown in Figure 4a. The average maximum load of CdTe QDs per 4 mg HPAMAM–DA was 1.26 mg, assuming the CdTe QDs have the same absorption coefficient both in water and in polymeric chloroform solution.

The absorption and photoluminescence spectra of CdTe QDs in the encapsulation and release process are displayed in Figure 5 and Figure 6, respectively. The absorption peaks of CdTe QDs changed from 521 nm to 508 nm after encapsulation and then shifted to 495 nm after release stage. A similar blue shift also happened to the emission peak of CdTe QDs. The emission peaks varied from 574 nm to 565 nm and to 545 nm for aqueous CdTe QDs, CdTe/HPAMAM–DA nanocomposites in a chloroform phase and CdTe/HPAMAM nanocomposites in an aqueous phase, respectively. This phenomenon can also be evidenced by the photograph taken under UV light, as shown in Figure 3b,d,f. Initially, the CdTe QDs had a yellow emission and exhibited a yellow-green emission after encapsulation. In addition, finally they had a green emission after releasing to aqueous phase. The blue shift of absorption and emission peak in the encapsulation process can be attributed to size selectivity effect of HPAMAM–DA on CdTe QDs and the complexation between CdTe QDs and HPAMAM–DA. In addition, the blue shift of absorption and emission peak in the release process should be ascribed to the etching of acid solution on CdTe QDs and the complexation between CdTe QDs and HPAMAM. For the photoluminescence spectra of CdTe/HPAMAM–DA nanocomposites and CdTe/HPAMAM nanocomposites, there is another emission peak at 428 nm, which comes from the emission of HPAMAM.

The absorption spectra of Au NPs in the encapsulation and release process are given in Figure 7. The absorption peaks of Au NPs encapsulated by HPAMAM–DA (Au/HPAMAM–DA) and released Au NPs in the form of Au/HPAMAM are at 532 nm and 527 nm, respectively. The Au NPs encapsulated by HPAMAM–DA have an claret-red colour in a chloroform phase, as shown in Figure 7a. Au NPs were then released to aqueous phase by adding PAA/HCl aqueous solution (pH 5.4) to break imine bonds of HPAMAM–DA (Figure 7b). During the imine hydrolysis, pH-responsive shell of HPAMAM–DA was cleaved and the remained water-soluable Au/HPAMAM nanocomposites transfered to aqueous phase. The shell cleavage of HPAMAM–DA was confirmed by IR (from Appendix A) and ^1^H NMR spectra (from Appendix A). The IR band of the –CHO peak at 1712 cm^−1^ appeared for the remained chloroform phase because of imine hydrolysis. While the NH_2_ bending vibration at 1651 cm^-1^ appeared for the final aqueous phase, implying that HPAMAM or HPAMAM–DA with partial hydrolysis were released to aqueous phase. The proton signals at 1.420–3.727 ppm in Appendix A can be assigned to HPAMAM and the extremely weak proton signal at 8.015 ppm (N=C*H*) indicates that the imines of HPAMAM–DA molecules were almost totally hydrolyzed. Consequently, we can draw the conclusion that HPAMAM molecules were released to aqueous phase after imine hydrolysis.

The release of nanocrystals from oil phase to water phase is a rather challenging subject compared with single water phase release. The imines can be cleaved at pH 7 in water phase, but they cannot be cleaved pH 7 in chloroform. More acidic aqueous solution is needed to destroy the imines in chloroform, but the NCs encapusulated should not be decomposed by acidic aqueous solution. PAA can ionize hydrogen protons continuously. That is why we use PAA to supply more hydrogen protons. HPAMAM-g-MPEG with acylhydrazones in chloroform solution actually could be used to encapsulate CdTe QDs from our experiment confirmation, but they cannot be used for CdTe QD release [46]. Because only aqueous acid with about pH ≤ 3 can completely destroy the acylhydrazones of HPAMAM-g-MPEG in chloroform, while CdTe QDs will be decomposed completely if aqueous acid with pH ≤ 3 was used. That is why we use covalent dynamic hyperbranched polymers with imine bonds not acylhydrazones to encapsulate and release NCs.

Transmission electron microscopy (TEM) images of CdTe QDs (Figure 8a–c) show that the monodispersity and average size of CdTe QDs did not significally change during the encapsulation and release stages. The average sizes of CdTe QDs in all stages were all about 2.6 nm. HPAMAM–DA, acted as unimolecular micelle, could segregate CdTe QDs in the unimolecular micelle and complexed with the QDs by its HPAMAM core, thus the QDs appeared to be monodisperse in the TEM grids. When the imine linkage in the HPAMAM–DA was broken and the remained water-soluble CdTe/HPAMAM nanocomposites would be released to aqueous phase. The relevant EDS spectrum shown in Figure 8d also proves that CdTe QDs have been released to aqueous phase. TEM images of Au NPs encapsulated by HPAMAM–DA in a chloroform phase and released Au NPs in an aqueous phase are shown in Figure 8e,f. The average size of Au NPs encapsulated by HPAMAM–DA in a chloroform phase and released Au NPs in an aqueous phase were all about 9.9 nm.

The hydrodynamic diameters of CdTe QDs and Au NPs in the encapsulation and release stages were characterized by DLS. The average diameters of pure CdTe QDs, HPAMAM and HPAMAM–DA as given in Figure 9a were 2.7, 3.2 and 10.1 nm, respectively. After encapsulating aqueous CdTe QDs by HPAMAM–DA, the diameter of the resulting CdTe/HPAMAM–DA nanocomposites had no distinct change as compared with that of HPAMAM–DA. This was because the extremely small CdTe QDs were encapsulated into the hydrophilic part of amphiphilic HPAMAM–DA by the complex interactions between HPAMAM and QDs. When the CdTe QDs were released into aqueous phase in the form of CdTe/HPAMAM nanocomposites, the diameter decreased to 4.8 nm due to the breakaway of hydrophobic alphatic chains on HPAMAM–DA. Figure 9b shows the hydrodynamic diameters of Au/HPAMAM–DA and Au/HPAMAM nanocomposites. When HPAMAM–DA amphiphilic hyperbranched polymers encapsulated Au NPs, their hydrodynamic diameter increased to 13.5 nm due to the large diameter of Au NPs as shown in Figure 8e. One Au NP was encapsulated per HPAMAM–DA unimolecular micelle, as estimated by DLS and TEM data. After Au NPs were released to aqueous phase, the resulting Au/HPAMAM nanocomposites had a diameter of 10.2 nm due to the cleavage of imines in HPAMAM–DA.

Bhatia et al. found that cadmium-based QDs such as CdSe-core QDs were acutely toxic under certain conditions, and free Cd^2+^ ions could be liberated from QDs, resulting the cytotoxicity of QDs [47]. While HPAMAM has been proved to have low cytotoxicity and high transfection efficiency [48]; thus, HPAMAM could be used to coat cadmium-based QDs and to decrease the cytotoxicity of cadmium-based QDs. Herein, the cytotoxicity of CdTe/HPAMAM nanocomposites and HPAMAM was evaluated by MTT assay in the Hela cell after 24 h culture. The cell viability after incubation with HPAMAM and CdTe/HPAMAM at concentrations ranging from 1 to 100 µg/mL are displayed in Figure 10. HPAMAM exhibits low toxicity, which might be related to its degradability and low charge density properties [48]. The cytotoxicity of CdTe/HPAMAM nanocomposites increase slightly compared with that of HPAMAM, which might be related to the toxicity of CdTe QDs.

Green et al. reported that bare cysteine-capped particles cannot be endocytosed by the cells and cationic liposome treated QDs can be internalized into human breast cancer cells [49]. Bhatia et al. reported that hepatocytes were then labeled by endocytosis of epidermal growth factor-conjugated QDs [47]. Thus, it can be seen that the transfection reagent is essential for cell imaging application. Benefiting from the properties of QDs and HPAMAM, the released CdTe/HPAMAM should be able to be applied in many biomedical fields. To assess the potential application of CdTe/HPAMAM as a bioimaging probe, Hela cells were incubated in a medium containing CdTe/HPAMAM for 6 h before characterization by laser scanning confocal microscope. Green fluorescence can be seen in the Hela cells shown in Figure 11b, indicating the CdTe/HPAMAM nanocomposites were internalized through endocytosis. The CdTe/HPAMAM nanocomposites integrate the advantages of HPAMAM and CdTe QDs; thus, the fluorescent CdTe/HPAMAM nanocomposites would also be a promising fluorescence probe for tracking drug release, gene transfection, etc.

Metal NPs such as Au, Pd NPs have been widely used as catalysts owing to their size and shape-dependent catalytic activities. Herein, catalytic activity of Au/HPAMAM nanocomposites was assessed by PNP hydrogenation catalysis, as illustrated in Scheme 1g. The PNP hydrogenation reaction was followed via UV–Vis spectra. From Figure 11c, we can see that the absorbance at 400 nm decreased, accompanied by an increase of the absorbance at 310 nm (peak of PAP), indicating PNP was being reacted into PAP under Au NP catalysis. The rate constants *k*_app_ were determined based on the first-order rate law [50,51]. By plotting the natural log of the corrected absorbance at 400 nm against time, and fitting the steepest part of the curve into a line, the negative slope was gained and recognized as the rate constant. The rate constant *k*_app_ for Au/HPAMAM nanocomposites was 0.136 s^−1^, as shown in Figure 11d. After being normalized to the surface area per unit volume, the rate constant *k*_1_ was 1.56 L·s^−1^·m^−2^, which was a relatively high value for the reduction of PNP [50,51,52,53,54]. G4 or G6 PAMAM dendrimer often has steric crowding problems at the periphery, while HPAMAM with about three thousand molecular weight has little steric effect for PNP to access the surfaces of Au NPs. That is why Au NPs passivated by HPAMAM have high rate constants. For the control experiment shown in Figure 12, when pure HPAMAM without Au NPs was added, the UV–Vis spectra of PNP had no conspicuous changes in 30 s and the reduction rate constant was nearly zero, implying that HPAMAM had no catalytic effect on PNP hydrogenation.

## 4. Conclusions

In this paper, HPAMAM–DA covalent dynamic hyperbranched polymers containing imine linkage were used to encapsulate and release CdTe QDs and Au NPs. Take CdTe QDs as an example, aqueous CdTe QDs were firstly phase transferred to chloroform phase by HPAMAM–DA and then released to aqueous phase by imine cleavage. By this strategy, the phase transfer between organic phase and aqueous phase for CdTe QDs can be readily realized, and application of the resulting CdTe QDs could be extended to not only optoelectronics but also biomedical field (such as bio-imaging, gene transfection). Furthermore, the optical properties of CdTe QDs can be tuned in different phase as the structure of HPAMAM–DA change along with imine cleavage.

The resulting CdTe/HPAMAM nanocomposites could easily be endocytosed by the Hela cells without transfection reagent and exhibited excellent biological imaging behavior. In addition, the resulting Au/HPAMAM nanocomposites were proved to have high catalytic activities on PNP hydrogenation.

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
