# Peer review of "Nanocrystal Encapsulation, Release and Application Based on pH-Sensitive Covalent Dynamic Hyperbranched Polymers"

_polymers, 2019, doi:10.3390/polym11121926_

Round 1
Reviewer 1 Report
The paper entitled “Nanocrystal encapsulation, release and application based on Ph-sensitive covalent dynamic hyperbranched polymers” by Shi Y. et al. has been reviewed. The paper deals with the elaboration of hyperbranched polymers with multi-arm bydrophobic chains and hydrophilic core. The experimental results seem to be very promising. But there are some remarks which should be taken into consideration before publication. To begin with, there are some mistakes in the English language (for example, line 284; line 286, and so on).1. Lines 186-191, Fig. 3b: why linear behavior of the trend lines was chosen? 2. Lines 212-216, Fig. 4 and 5: for better understanding the authors should add the values of wavelength to Figures.3. Fig. S4 and Fig. S5 should be added in the text.
Reviewer 2 Report
The authors described the synthesis of a dendrimer system able to form complexes with CdTe and Au-NP. These systems can be utilized for bio-imaging.
The manuscript cannot be published in its current form and, in particular, the novelty and the gained information is questionable.
In detail:
1.) The authors utilized in several studies before a similar research approach. The only novelty is the utilization of imines as pH-triggerable unit. However, acylhydrazones (esters) were utilized before, which also works. Thus, I see no benefit of using imines. There is also no comparison to literature of the current results. Furthermore, the advantage of this study is not presented at all. What is new?
2.) There are a lot of studies describing Au-NP (also QD) in dendrimers. What is the benefit compared to all those? What can we learn from your approach?
3.) The materials and the synthesis is not described in an ideal manner, e.g., poly(acrylic acid) should be characterized. The IR is missing at all (in the materials section). The NMR values cannot be determined in such an exact manner, in particular in polymers.
4.) The IR spectra in Figure S1: I cannot follow the conclusions. The spectra are too similar to draw any conclusion. The error of such IR-spectra is around 2-5 cm-1 and you interpretate shifts of 1 cm-1. This is no proof of successful functionalization.
5.) Furthermore, the NMR is also not sufficient enough for proving the functionalization. Did you integrate the signals? Does the "signal" at 7.9 ppm fits to the alkyl chains of the dodecyl? Did you measure a DOSY NMR showing that the diffusion coeffient is equal for all signals?
6.) Did you measure SEC to show that the molar mass or at least the hydrodynamic radius changed during functionalization?
7.) Did you test encapsulation without those imine groups? Is there a different?
8.) For the cyctotoxicity: Did you test HPAMAM with and without the imino groups? Any difference?
9.) The conclusion must be improved and it should be expanded by a section what the major information of the manuscript is.
In general: The current study seems to be a slight variation of all studies before without any significant news. Presumably, the system is not even better than before and no information can be obtained by reading the manuscript.
Reviewer 3 Report
My review has been attached.

Round 2
Reviewer 2 Report
The authors only partially answered the questions. Further points have to be answered:
SEC values are provided in a too exact manner. SEC is a relative methods a values of 3848 g/mol cannot be detected. I suggest to use: 3800 g/mol instead. By the way: How is it possible that the molar mass of a substance changes during revision? Did you repeat the experiment, did you use other polymer? The justification for the novelty should not only be provided in the point to point answer. It should somehow added to the manuscript. For the cyctotoxicity: The question is not answered at all. I would like to known: Does the addition of imines into the structure (compared to the same system without) changes the cyctotoxicity? The agrumentation about amines is useless in that case.
